# Targeting Endothelial Necroptosis Disrupts Profibrotic Endothelial–Hepatic Stellate Cells Crosstalk to Alleviate Liver Fibrosis in Nonalcoholic Steatohepatitis

**DOI:** 10.3390/ijms241411313

**Published:** 2023-07-11

**Authors:** Mengli Yan, Hui Li, Shiyu Xu, Jinyan Wu, Jiachen Li, Chengju Xiao, Chunheng Mo, Bi-Sen Ding

**Affiliations:** 1Key Laboratory of Birth Defects and Related Diseases of Women and Children of MOE, State Key Laboratory of Biotherapy, West China Second University Hospital, Sichuan University, Chengdu 610064, China; yanmengli2020@163.com (M.Y.); echo_lee21@163.com (H.L.); shiyuxu@stu.csu.edu.cn (S.X.); 15892682656@163.com (J.W.); hansodeity@163.com (J.L.); cj_xiao@yeah.net (C.X.); 2Fibrosis Research Program, Division of Pulmonary and Critical Care Medicine, Division of Liver Diseases, Icahn School of Medicine at Mount Sinai, New York, NY 10029, USA; 3Division of Regenerative Medicine, Weill Cornell Medicine, New York, NY 10065, USA

**Keywords:** liver fibrosis, nonalcoholic steatohepatitis (NASH), necroptosis, endothelial cells, mixed lineage kinase domain-like protein (MLKL), endothelial–HSC crosstalk

## Abstract

Chronic liver diseases affect over a billion people worldwide and often lead to fibrosis. Nonalcoholic steatohepatitis (NASH), a disease paralleling a worldwide surge in metabolic syndromes, is characterized by liver fibrosis, and its pathogenesis remains largely unknown, with no effective treatment available. Necroptosis has been implicated in liver fibrosis pathogenesis. However, there is a lack of research on necroptosis specific to certain cell types, particularly the vascular system, in the context of liver fibrosis and NASH. Here, we employed a mouse model of NASH in combination with inducible gene knockout mice to investigate the role of endothelial necroptosis in NASH progression. We found that endothelial cell (EC)-specific knockout of mixed lineage kinase domain-like protein (MLKL), a critical executioner involved in the disruption of cell membranes during necroptosis, alleviated liver fibrosis in the mouse NASH model. Mechanistically, EC-specific deletion of *Mlkl* mitigated the activation of TGFβ/Smad 2/3 pathway, disrupting the pro-fibrotic crosstalk between endothelial cells and hepatic stellate cells (HSCs). Our findings highlight endothelial MLKL as a promising molecular target for developing therapeutic interventions for NASH.

## 1. Introduction

Non-alcoholic fatty liver disease (NAFLD) is a prevalent condition in the world [1] characterized by the accumulation of fat in the livers of individuals who do not consume substantial amounts of alcohol [2]. NAFLD is considered a significant public health concern due to its potential to progress to more severe liver damage, including non-alcoholic steatohepatitis (NASH), cirrhosis, and hepatocellular carcinoma (HCC) [3]. NASH is a more severe form of NAFLD characterized by ballooned hepatocytes along with steatosis and lobular inflammation, leading to liver fibrosis [4]. The incidence of non-alcoholic fatty liver disease (NAFLD) in China has been increasing year by year. According to a nationwide epidemiological survey, the prevalence of NAFLD among Chinese adults has exceeded 20%, and the rate is over 60% in the obese population [5,6]. It is estimated that approximately 20–30% of patients with NAFLD will progress to NASH [4]. The prevalence of NASH in the Chinese population ranges from 2.4 to 6.1% [7,8,9,10,11], which is close to the global prevalence of NASH (3.0–5.0%) [7,12]. Among them, the prevalence of NASH is higher in Chinese men (6.89%) than in women (5.04%) [8]. Over the past decade, with changes in lifestyle, the incidence of NASH in China has significantly increased and has become a major public health issue [13]. NASH is currently the second leading indication for liver transplantation in the United States and is associated with a significant economic burden [14,15]. Its clinical significance and impact on healthcare systems worldwide have drawn significant attention in recent years, since there is currently no effective treatment for NASH [15]. Therefore, understanding NASH pathogenesis is critical to addressing this growing public health issue.

Necroptosis, a type of programmed cell death initiated by cellular stress, infection, or disease [16,17,18,19], is characterized by the activation of receptor-interacting protein kinases (RIPKs), particularly RIPK1 and RIPK3, which form a complex known as the necrosome [20,21,22]. The necrosome activates downstream effector proteins, including the mixed lineage kinase domain-like protein (MLKL). MLKL is phosphorylated and translocated to the plasma membrane. It promotes cell membrane permeabilization and the release of intracellular contents, ultimately leading to cell death [23,24,25,26,27]. The critical role of MLKL in the execution of necroptosis has been widely recognized. Preventing the activation or loss of function of MLKL can block necroptosis [21,28]. MLKL-mediated necroptosis contributes to the pathogenesis of liver disorders, such as drug-induced liver inflammation [24] and injury, and NASH [29,30]. NASH induced by high-fat feeding in mice is associated with the upregulation of MLKL expression, and the inhibition of necroptosis reduces inflammation linked with NASH, such as damage-associated molecular patterns (DAMPs) [29,31,32,33,34,35,36]. Pharmacological blockade of necroptosis ameliorates hepatic inflammation and fibrosis in diverse murine models of liver disorders [34,36].

The liver lobules are comprised of hepatocytes and various non-parenchymal cells (NPCs) [37]. Among these NPCs, liver sinusoidal endothelial cells (LSECs) make up 50% of the total population [37]. Through autocrine and paracrine functions, LSECs modulate the hepatocyte response to different insults and regulate neighboring cells such as hepatic stellate cells (HSCs) and immune cells [38,39,40,41]. Dysfunction of LSECs contributes to liver fibrosis and NASH progression [39,40,42,43,44]. Gonzalez et al. demonstrated that rats fed a high-fat diet exhibited decreased NO activity and increased oxidative stress, indicating LSECs’ dysfunction [45]. Peng et al. also found that LSECs morphology is damaged in NASH rats, with most LSECs showing capillarization and a reduction in fenestration [46]. Moreover, studies have shown that LSECs produce pro-inflammatory cytokines such as TNF-a, IL-6, IL-1, and CCL2 leading to maladaptive vascular niches in NASH [47]. Additionally, LSECs in the context of NASH have been shown to overexpress vascular adhesion proteins which promote the activation of HSCs and lead to liver fibrosis [48].

Currently, research on necroptosis in the fibrotic micro-environment focuses on fibroblasts, macrophages, and neutrophils [49,50,51]. It is widely accepted that necroptosis leads to plasmalemma rupture, organelle swelling, and the release of cell components, resulting in inflammation and activation of neighboring cells in response to changes in the micro-environment [21,27,33,52]. Endothelial cells, especially LSECs, play a critical role in maintaining homeostasis, regulating body metabolism, facilitating tissue repair, and directing organ regeneration [41,53,54,55,56,57]. In response to organ damage, angiocrine factors promote the self-renewal of tissue-specific stem and progenitor cells and the differentiation of those cells into functional organs [58]. The role of endothelial necroptosis in NASH remains unclear. In our study, we will generate endothelial cell-specific knockout of *Mlkl* gene mice to investigate the role of endothelial necroptosis in NASH. This study aims to provide valuable insights into NASH pathogenesis and open new avenues for developing effective treatments by targeting endothelial necroptosis.

## 2. Results

### 2.1. Construction of Inducible Endothelial Cell (EC)-Specific Knockout of Mlkl Mice (Mlkl^iΔEC/iΔEC^)

Necroptosis has been implicated in liver fibrosis, yet there is a lack of research on necroptosis specific to certain cell types, particularly within the vascular system, in the context of liver fibrosis and NASH. To examine whether necroptosis of endothelial cells is involved in NASH and the fibrosis model, we generated a conditional *Mlkl* knockout mouse model using CRISPR/Cas9 technology. A gRNA targeting sequence of the mouse *Mlkl* gene following the PAM site was inserted into the Exon3 locus and introduced into mice (Figure 1A). Genotyping assays were performed to distinguish homozygote (*Mlkl*^f/f^), heterozygote (*Mlkl*^f/+^), and wild-type (WT) mice (Figure 1B top). We also screened mice carrying VE-cad-Cre^ERT2^ to subsequently breed conditional knockout mice (Figure 1B bottom). Mice carrying EC-specific promoter Cdh5/VE-cadherin-driven tamoxifen-responsive Cre (VE-cad-Cre^ERT2^) were crossed with *Mlkl*-floxed mice. The injection of tamoxifen into the resulting mouse offspring selectively induced deletion of *Mlkl* in vascular endothelial cells, generating EC-specific knockout of *Mlkl* in adult mice (*Mlkl*^iΔEC/iΔEC^). Mice expressing Cdh5-PAC-Cre^ERT2^ alone served as the control group (Figure 1C). After inducible deletion of *Mlkl* by tamoxifen injections, ECs were isolated from liver tissues. RT-qPCR showed that *Mlkl* relative mRNA levels in liver ECs of *Mlkl*^iΔEC/iΔEC^ mice dropped to 25% compared with *Mlkl*^+/+^ mice (Figure 1D). In parallel, we also examined knockout efficiency after tamoxifen injection, as assessed by the protein levels of MLKL (Figure 1E). MLKL protein levels in liver ECs were reduced significantly. The result of quantification by image J was consistent with Western blot (Figure 1F). These data indicate that construction of EC-specific knockout of *Mlkl* mice is successful.

### 2.2. EC-Specific Knockout of Mlkl Alleviates Histopathological Phenotype Associated with NASH Progression

To explore the influence of necroptosis of ECs on liver fibrosis in NASH model, repetitive CCl_4_ injection and a high fat and sugar diet (WD diet) were used to induce the NASH model (Figure 2A). One dose of CCl_4_ per week was injected into the enterocoelia of mice, and liver functions, steatosis, inflammation, liver fibrosis, collagen deposition, and index of liver fibrosis were examined after 12 weeks. Firstly, liver functions of the control and *Mlkl*^iΔEC/iΔEC^ mice were assessed. We found that the liver functions of genetic knockout of *Mlkl* mice was improved. Specifically, the serum concentration of aspartate amino transferase (AST), alanine aminotransferase (ALT), and alkaline phosphatase (ALP) were decreased in *Mlkl*^iΔEC/iΔEC^ mice compared with the control group (Figure 2B). Steatosis is a major pathological feature of NASH [59]. Therefore, we performed Oil Red O staining and found that lipid droplet deposition in the livers of the control group was higher than that large surface area (Figure 2C). The change in serum cholesterol (CHO) in *Mlkl*^iΔEC/iΔEC^ mice was consistent with the above results (Figure 2B). Next, we assessed the histopathology and collagen deposition (Figure 2D, right and 2E) in the livers of the control and *Mlkl*^iΔEC/iΔEC^ mice. Compared with the control group, which exhibited prominent histological features of NASH, including inflammation and ballooned hepatocytes (as indicated by black arrows), the livers of *Mlkl*^iΔEC/iΔEC^ animals showed an improved phenotype (Figure 2D left).

The excessive accumulation of extracellular matrix (ECM) proteins, including collagen (especially Collagen I and III), is a prominent hallmark of liver fibrosis. In addition, α-smooth muscle actin (α-SMA) serves as a marker of pathologic fibroblasts in fibrotic tissue. Therefore, to further assess the molecular characteristics of fibrosis, total RNA was isolated from the livers of *Mlkl*^iΔEC/iΔEC^ and *Mlkl*^+/+^ NASH mice and RT-qPCR was performed. The mRNA expression levels of *Col1a1*, *Col3al*, and *α-SMA* (*Acta2*) were upregulated in *Mlkl*^+/+^ mice compared with *Mlkl*^iΔEC/iΔEC^ mice (Figure 3A). Meanwhile, the liver hydroxyproline (HYP) abundance, one of the main components of collagen, also significantly reduced in the NASH model with endothelial *Mlkl* deletion (Figure 3B). Consistently, we also examined the fibrotic responses in the NASH model, as assessed by the protein levels of α-SMA, Desmin, and Collagen I (Figure 3C,D). The protein levels of α-SMA, Desmin, and Collagen I were reduced in the genetic deletion of *Mlkl* mice. Analysis of immunostained liver sections obtained from individuals with NASH also suggested downregulation of Collagen I (Figure 3E), α-SMA (Figure 3F), and Desmin (Figure 3G) deposition compared to the corresponding expression in control mice livers (Figure 3H). Collectively, these findings suggest that the fibrotic response in *Mlkl*^iΔEC/iΔEC^ mice is downregulated in the NASH model, indicating that MLKL plays an important role in the progression of liver fibrosis.

### 2.3. EC-Specific Knockout of Mlkl Downregulates the Liver Fibrosis Indexes after Repeated CCl_4_ Injury

To examine whether the effects of necroptosis of ECs in the NASH model can be generalized to another liver injury model, we used repeated CCl_4_ injection to investigate the contribution of vascular maladaptation to liver fibrosis and the underlying mechanisms (Figure 4A). According to our previous research, both control mice and *Mlkl*^iΔEC/iΔEC^ mice were treated with 40% CCl_4_ every 3 days to establish a chronic liver fibrosis model [60]. After 3 weeks, mice were sacrificed and liver tissues were harvested to evaluate the fibrotic responses. Since lymphatic vessel endothelial receptor-1 (Lyve-1) is the marker of ECs, endothelial damage after injury was tested by immunostaining of the ECs markers Lyve-1 (Figure 4B). The staining results suggested a liver vascular maladaptation in this mouse liver injury model. We found that, in the *Mlkl*^iΔEC/iΔEC^ group, liver function parameters of mice, including ALT, ALP, and total bilirubin (TBIL), were lower than those in the control group (Figure 4C). As explicated in Figure 4D, there were large contiguous fibrotic masses in the H&E staining liver sections in the control group. Furthermore, the liver architecture was severely damaged, and part of it was not preserved. In the Sirius red staining results, collagen fibers were dyed bright yellow or red. It showed that the mean staining intensity of collagen deposition is about 50% of the microscopic field. However, after knockout of *Mlkl* in mouse liver ECs, the damage of the liver tissue was reduced. There was no doubt that knockout of *Mlkl* in liver ECs could reduce collagen fibers deposition compared to the control group, and the liver sections of the *Mlkl*^iΔEC/iΔEC^ group tended to be normal.

Next, we found that in the CCl_4_-induced fibrosis model, the mRNA expression of *Col1a1*, *Col3a1*, and *α-SMA* increase, while these elevations were abrogated in the *Mlkl*^iΔEC/iΔEC^ group, as shown by RT-qPCR (Figure 5A). Similarly, we also found that the level of hydroxyproline was increased after liver injury but reduced in the *Mlkl*^iΔEC/iΔEC^ group. In parallel, we examined fibrotic responses after injection of CCl_4_, the protein levels of α-SMA and Collagen I were downregulated in the *Mlkl*^iΔEC/iΔEC^ group, compared to the control group (Figure 5C,D). Immunofluorescence results, as quantified in Figure 5E–G, verified that knockout of *Mlkl* in mouse liver ECs restrained the upregulation of Collagen I and Desmin induced by CCl_4_ injection. Based on these findings, we postulated that necroptosis of ECs might contribute to impaired liver repair and fibrosis after injury. Therefore, targeting of endothelial MLKL during chronic liver injury could potentially alleviate fibrosis.

### 2.4. EC-Specific Knockout of Mlkl Reduces the Pro-Fibrotic TGFβ/Smad 2/3 Signaling Axis between Endothelial Cells and HSCs

Based on the significant attenuation of liver fibrosis in NASH mice with EC-specific knockout of *Mlkl*, we sought to define the molecular mechanisms underlying the anti-fibrotic effect of *Mlkl^i^*^ΔEC/iΔEC^ mice. The TGFβ/Smad2/3 pathway is a major profibrogenic signal for HSCs activation and induction of the fibrotic response [61]. Primary endothelial cells were isolated from livers of indicated mice, and the RNA extracting from harvested ECs in Figure 6A was used for RT-qPCR. We found that the expression of profibrotic gene TGFβ was decreased in the NASH model of *Mlkl*^iΔEC/iΔEC^ mice (Figure 6B). To determine the effects of endothelial MLKL on the TGFβ/Smad 2/3 signaling in HSCs, we examined the protein levels of the phosphorylation of Smad2/3 (p-Smad2/3) in corresponding experimental groups. Immunoblotting showed that p-Smad2/3 levels were reduced in the genetic deletion of *Mlkl* mice (Figure 6C). We also performed double immunofluorescence staining of p-Smad2/3 with α-SMA (an activated HSC marker) [48]. In the *Mlkl^+/+^* NASH mice, we observed strong colocalization of p-Smad2/3 and α-SMA (white arrow in Figure 6D). However, the proportion of colocalization was dramatically reduced in the genetic deletion of *Mlkl* mice (Figure 6D), indicating that necroptosis of endothelial cells might regulate the fibrosis response via TGFβ/Smad 2/3 signaling axis. Collectively, these findings indicate that EC-specific deletion of *Mlkl* effectively alleviates liver fibrosis in NASH by suppressing the activation of the TGFβ/Smad 2/3 pathway and disrupting the pro-fibrotic crosstalk between endothelial cells and HSCs.

## 3. Discussion

The pathogenesis of NASH is a complex process and its cellular and molecular mechanisms remain unclear, which hinders the development of therapeutic strategies for NASH [62,63,64]. In the present study, we showed that endothelial necroptosis contributes to NASH progression and blockage of endothelial necroptosis alleviates liver fibrosis in NASH. NASH development and progression have been linked to different forms of cell death, including necroptosis, ferroptosis, pyroptosis, and apoptosis [65,66,67,68]. In addition to necroptosis of endothelial cells in our study, there is emerging evidence that necroptosis in other cell types may play a role in NASH pathogenesis and modulation of necroptosis in other cell types has been shown to alleviate liver fibrosis and NASH progression in experimental models [69,70]. A potential therapeutic approach for biliary atresia (BA) might be to target macrophage necroptosis, which plays an important role in BA liver fibrosis pathogenesis [71]. It is reported that necroptotic hepatocytes can trigger the recruitment of immune cells, leading to inflammation and contributing to NASH development and progression. Clearance of necroptotic hepatocytes by liver macrophages is a potential strategy to mitigate hepatic stellate activation and dampen liver fibrosis in NASH [72,73,74,75]. Hepatic stellate cell necroptosis induced by curcumol might provide a therapeutic strategy for the treatment of liver fibrosis [76]. Overall, these findings suggest that targeting necroptosis in various cell types may represent an effective therapeutic strategy to treat liver fibrosis and dampen NASH progression.

Our study suggests that endothelial necroptosis contributes to liver sinusoidal endothelial cell loss and dysfunction, thereby resulting in NASH development and progression. Capillaries are lined with endothelial cells, which act as a barrier between the circulatory system and tissues [77,78]. As part of blood vessels, endothelial cells participate in delivering blood and nutrients. Vascular endothelial cells can also establish vascular niches by deploying various angiocrine factors such as TGFβ, hepatocyte growth factor (HGF), and Wnt signaling molecules to guide a variety of physiological processes, including liver regeneration, fibrosis, and NASH progression [41,57,79]. Therefore, it is possible that as a result of necroptosis, endothelial cells become lost and dysfunctional, leading to maladaptive vascular niches causing liver fibrosis in NASH.

Liver sinusoidal endothelial cells (LSECs) can influence the occurrence and development of NASH by regulating adjacent hepatic stellate cells (HSCs) and immune cells [38,39,42,80]. Deleve et al. have shown that LSECs can prevent HSC activation and fibrosis in physiological conditions [81]. However, in pathological conditions, LSECs promote the activation of HSCs and macrophages by secreting pro-fibrotic and pro-inflammatory factors, leading to liver fibrosis [82,83,84]. Activated Kupffer cells promote LSEC capillarization, thereby regulating the hepatic microenvironment by directly secreting ECM and pro-fibrotic factors, promoting the progression of steatosis to NASH [44,85,86,87,88,89,90]. In this study, we found that the necroptosis of endothelial cells promotes TGFβ expression and activates HSCs, thereby promoting liver fibrosis in NASH. TGFβ is a vital profibrogenic cytokine for HSCs activation, which is the dominant event in hepatic fibrogenesis. Mechanically, TGFβ binds to its receptor (TβRI, TβRII) and leads to the phosphorylation of Smad2/3 [61]. The p-Smad2/3 is required for their nuclear translocation and transcriptional regulation of their fibrotic target genes (Figure 7) [61]. Our results indicate that necroptosis of endothelial cells might produce a pro-fibrotic endothelial signal involving TGFβ which activates Smad2/3 signaling in HSCs. Indeed, EC-specific deletion of *Mlkl* blocked necroptosis and reduced the activation of the TGFβ/Smad 2/3 pathway, thereby blocking a pro-fibrotic communication between endothelial cells and HSCs (Figure 7). Targeting of the communication between endothelial cells and HSCs could potentially facilitate therapy for liver fibrosis and NASH. Guo et al. have shown that Breviscapine, a natural flavonoid prescription drug isolated from the traditional Chinese herb Erigeron breviscapus, alleviates NASH through reducing lipid accumulation, pro-fibrotic factors, and inflammation in hepatocytes [91]. LSECs and hepatocytes are the two most abundant cell types in the liver, with LSECs accounting for 15%–20% of the total cells in livers [92]. Dysfunctional hepatocytes and endothelial cells may primarily regulate the occurrence and development of NASH through synergistic effects. Therefore, developing approaches that target dysfunctional hepatocytes and endothelial cells in a coordinated manner to reduce pro-fibrotic factors and inhibit inflammation and HSC activation may be an effective therapeutic strategy for NASH.

Currently, there is no approved drug for the treatment of NASH, and it is considered an area of highly unmet medical need [93,94]. The global NASH drug market is expected to exceed USD 10 billion by 2025 [95]. Several drugs targeting specific molecular pathway are currently in clinical trials to mitigate NASH [93,96,97,98]. MLKL represents a critical factor in the process of necroptosis [18,99]. In the present study, we found that EC-specific knockout of MLKL effectively alleviates the progression of NASH. Building upon this insight, researchers can design inhibitors that specifically target MLKL in endothelial cells, or develop specific adeno-associated viruses (AAV) to knock down MLKL in endothelial cells. This approach achieves the objective of specifically blocking endothelial necroptosis, repairing the vascular microenvironment, and mitigating liver fibrosis in NASH.

## 4. Materials and Methods

### 4.1. Mice

C57BL/6J mice were obtained from GemPharmatech Co., Ltd. (Nanjing, China) and C57BL/6JSmoc-*Mlkl^em1(flox)Smoc^* mice were acquired from Shanghai Model Organisms Co., Ltd. (Shanghai, China). All mice were maintained in a pathogen-free facility at the Experimental Animal Center of West China Second Hospital, and were fed freely on a standard 12-h light–dark cycle. Animal experiments were approved by the Experimental Animal Ethics Committee of West China Second Hospital of Sichuan University. All experimental mice were randomly allocated to each group.

The inducible *Mlkl* knockout mice with *loxP*-flox fragments were generated through homologous recombination of the fertilized egg using CRISPR/Cas9 technology. Specifically, the flox region of the coding sequence (CDS) region of *Mlkl* was replaced, and the recombinant site was on chromosome 8 (ENSMUSG00000012519.15). According to the Cre-loxP system, tamoxifen-induced activation of Cre recombinase can effectively delete the sequence between two loxP sites, achieving specific knockout of the *Mlkl* gene. To achieve endothelial cell-specific gene knockout, mice carrying endothelial cell (EC)-specific promoter Cdh5/VE-cadherin-driven tamoxifen-responsive Cre (VE-cad-CreERT2) were crossed with Mlkl-floxed mice. Mice were injected intraperitoneally with 2% tamoxifen (Sigma-Aldrich, Burlington, MA, USA; cat: T5648) dissolved in corn oil (Sigma), which were treated once a day for three days, and then treated again at intervals of three days, for a total of nine times. The dosage is 100 mg/kg body weight (calculated by mouse weight), as previously described [60,100].

### 4.2. Genotyping

To obtain DNA for genotyping, 1–2 mm tissues were removed from the toe or tail tip of mice and mixed with 100 μL tissue lysis solution, which contained 1 x MGB buffer, 10% Triton, 1% β-Mercaptoethanol, premixed with protease K (20μL protease K/mL lysis solution; Sigma-Aldrich; cat: 39450-01-6). The mixture was incubated at 56 °C overnight. After denaturation, the samples were stored at 4 °C as a DNA template for genotyping. For the detection of *Mlkl* and Cre, 2× Taq Plus MasterMix with dye (Cwbio, Beijing, China; cat: CW2849) was mixed with the related primers, and polymerase chain reactions (PCR) were performed. Finally, the PCR products were subjected to agarose gel electrophoresis to determine the genotype, as previously described [41,60].

### 4.3. NASH Model

NASH model was induced by a Western diet (WD) and chemical injury [101]. After the mice were treated with tamoxifen for three weeks, they were fed with a high-fat diet containing 21.1% fat, 41% sucrose, and 1.25% cholesterol, in a weight ratio, along with a high sugar aqueous solution diet (23.1 g/L fructose and 18.9 g/L glucose) for 12 weeks. In addition, 20% carbon tetrachloride (CCl_4_) in corn oil was injected intraperitoneally once a week for 12 weeks, with a dosage of 2.5 mL/kg body weight (calculated by mouse weight). After 12 weeks of treatment, serum was collected from the fundus venous plexus, and the mice were sacrificed and perfused with 0.9% saline through the mice left ventricle. The liver lobes were then harvested for further testing, as previously described [63,102].

### 4.4. Chemically Induced Liver Fibrosis Model

Mice were fed with a standard diet for at least three weeks after the induction of the tamoxifen treatment. Mice were then injected intraperitoneally with 40% carbon tetrachloride/corn oil solution every three days in a total of three times [41]. Control mice were treated with the same dose of corn oil for the same duration. After three times of treatment, mouse serum was obtained through the fundus venous plexus, and the mice were sacrificed and perfused with 0.9% saline through the left ventricle. The liver lobes were harvested for further testing, as previously described [60].

### 4.5. Blood Biochemistry

Blood samples were collected and kept at room temperature for 30 min. After centrifugation at 3000 g for 20 min at 4 °C, the supernatant was aspirated, and serum was isolated and stored at −80 °C for biochemical detection. The serum levels of aspartate transaminase (AST), alanine transaminase (ALT), ALP (alkaline phosphatase), total bilirubin (TBIL), and cholesterol (CHO) were measured by the automated biochemical analyzer (Roche, Beijing, China;) for animals, as previously described [94].

### 4.6. Histochemical Staining

Liver tissues were harvested for histological analysis flowing a previously described protocol [41,57]. In brief, after sacrificing the mice, a piece of liver from each mouse was fixed in 4% paraformaldehyde (PFA) overnight at 4 °C. The liver tissues were then dehydrated, embedded in paraffin wax, and sectioned into 5 μm slices. These slices were stained with hematoxylin and eosin (H&E) and Sirius red for morphological assessment. Frozen liver tissue sections (10μm) were stained with Oil Red O to evaluate histopathological changes under microscopy (3D Pannoramic MIDI, Budapest, Hungary).

### 4.7. Immunofluorescence

Fresh liver samples were embedded in Tissue-Tek^®^ O.C.T. Compound and stored at −80 °C. Liver cryosections (8 μm in thickness) were fixed in 4% paraformaldehyde; all sections were permeabilized in 0.3% Triton X-100 for 20 min and then blocked with 2% BSA for 10 min. The sections were incubated with primary antibodies against Collagen I (Abcam, Waltham, MA, USA; cat: ab34710), α-SMA (Abcam; cat: ab7817), Desmin (Abcam; cat: ab15200) and Lyve1 (Fitzgerald, Dublin, Ireland; cat: 70R-LR003) overnight at 4 °C. Corresponding fluorescent secondary antibodies conjugated with Alexa Fluor 488 (1:500; Jackson ImmunoResearch, West Grove, PA, USA) were used to stain the sections for 1 h at room temperature in the dark, after which the primary antibodies on the slides were washed away. The nuclei were stained with DAPI (Thermo Fisher Scientific, Waltham, MA, USA) for 5 min. The slides were imaged with a Zeiss LSM 900 microscope and analyzed with ImageJ software (v1.8.0, Institutes of Health (NIH) Bethesda, MD, USA), as previously described [102].

### 4.8. Isolation of Endothelial Cells from Liver Tissues

On the last day of NASH and CCl_4_-induced liver fibrosis modeling, mice were sacrificed and perfused with 0.9% saline through the left ventricle. Perfused liver tissues were minced and dissociated with digestive enzymes, including Collagenase Ⅰ (Gibco, Billings, MT, USA; cat: 17100-017), DNase Ⅰ (Sigma-Aldrich; cat: D4527), at 37 °C for 25 min. The digested cell suspension containing liver sinusoidal endothelial cells (LSECs) were incubated with Dynabeads magnetic beads (Invitrogen, Waltham, MA, USA; cat: 11035) coated with rat anti-mouse CD31 (BD Bioscience, San Jose, CA, USA; cat: 553370). Negative depletion with anti-CD45 (BD Bioscience; cat: 553076) Dynabeads were performed to exclude hematopoietic cells. The cells collected after all treatments were CD31^+^CD45^−^. The collected LSECs were stored at −80 °C for further experiments, as previously described [103].

### 4.9. RNA Extraction and Real-Time Quantitative PCR (qPCR)

Total RNA was extracted from the liver tissues or isolated liver endothelial cells using TRIzol^TM^ (Invitrogen), following the manufacturer’s instructions. RNA reverse transcription was performed using the PrimeScript^TM^ RT reagent Kit (Takara, San Jose, CA, USA; RR047A-1), according to the manufacturer’s instructions. Relative RNA expressions were detected using SYBR Green Master Mix (Vazyme, Nanjing, China; cat: Q712-02/03) on a CFX96 Real-Time PCR System (Bio-Rad, Hercules, CA, USA), as previously described [60].

### 4.10. Western Blotting

The liver tissue and the collected LSECs were lysed with RIPA buffer containing 1% PMSF and loaded into the wells of the SDS-PAGE gel. After transferring the gel containing proteins onto PVDF membranes, the immunoblots were incubated with the primary antibodies against GAPDH (Servicebio, Wuhan, China; GB11002), MLKL (Huabio; cat: ET1601-25), Smad2/3 (Cell Signaling Technology, Danvers, MA, USA; cat: 8685), and p-Smad2/3 (Cell Signaling Technology; cat: 8828) for more than 8 h at 4 °C. The corresponding HRP-linked secondary antibody was then added and incubated for 1 h. The PVDF membrane containing target protein was laid on the exposure plate for the detection of protein. ImageJ software was used for image analysis as previously described [103].

### 4.11. Hydroxyproline (HYP) Analysis

Hydroxyproline levels were detected using a HYP assay kit (Solarbio, Beijing, China; cat: BC0255). Liver tissues were cut into pieces and digested in the extraction solution at approximately 100 °C for 2–6 h, until the tissues were totally hydrolyzed. The PH of the mixture was adjusted to 10 using 10 M NaOH. The adjusted mixture was then centrifuged at 16,000 rpm for 30 min, and the supernatant was collected for testing. The OD value of each well was measured at 560 nm with a SpectraMax ABS PLUS reader. The HYP content was calculated as previously described [94].

### 4.12. Statistical Analysis

All data were expressed as mean ± SEM. Statistical analysis was performed using Student’s *t*-test for comparisons between two groups or one-way ANOVA for experiments with more than two groups, using GraphPad Prism 8 (GraphPad Software, San Diego, CA, USA). A *p* value < 0.05 was considered statistically significant.

## 5. Conclusions

In summary, our findings indicate that the occurrence of endothelial necroptosis during the progression of NASH exacerbates the pathological process. Notably, EC-specific ablation of MLKL effectively attenuates liver fibrosis in NASH by blocking necroptosis and mitigating the TGFβ/Smad 2/3 signaling axis between endothelial cells and HSCs. Our study contributes to a better understanding of the underlying mechanisms involved in NASH progression and provides novel therapeutic approaches for this disease.

## Figures and Tables

**Figure 1 ijms-24-11313-f001:**
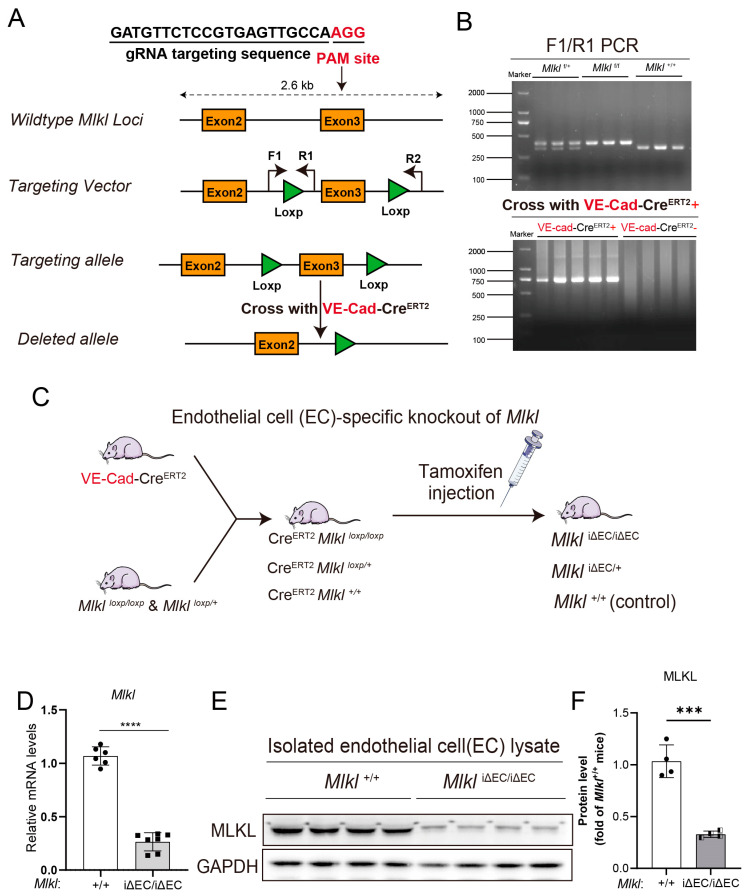
The construction of inducible endothelial cell-specific knockout of *Mlkl* (*Mlkl^i^*^ΔEC/iΔEC^) mice. (**A**) Schematic overview of generation of conditional *Mlkl* knockout mice using CRISPR/Cas9 system. F1: Forward primer 1, R1: Reverse primer 1, R2: Reverse primer 2. (**B**) The genotyping results of *Mlkl*^+/+^ (WT, wild type), *Mlkl*^f/f^ (*Mlkl^flox/flox^), Mlkl*^f/+^ (*Mlkl^flox/+^)*, VE-cad-Cre^ERT2+^, and VE-cad-Cre^ERT2-^ mice. (**C**) The breeding schema illustrating generation of endothelial cell (EC)-specific knockout of *Mlkl* in adult mice. (**D**) Inducible knockout of *Mlkl* in mouse liver endothelial cells were determined by RT-qPCR, Western blotting (**E**) and quantified by densitometry using image J and shown as fold of *Mlkl*^+/+^ mice (**F**). *n* ≥ 4 per group. Data are shown as means ± SEMs. *** *p* < 0.001, **** *p* < 0.0001.

**Figure 2 ijms-24-11313-f002:**
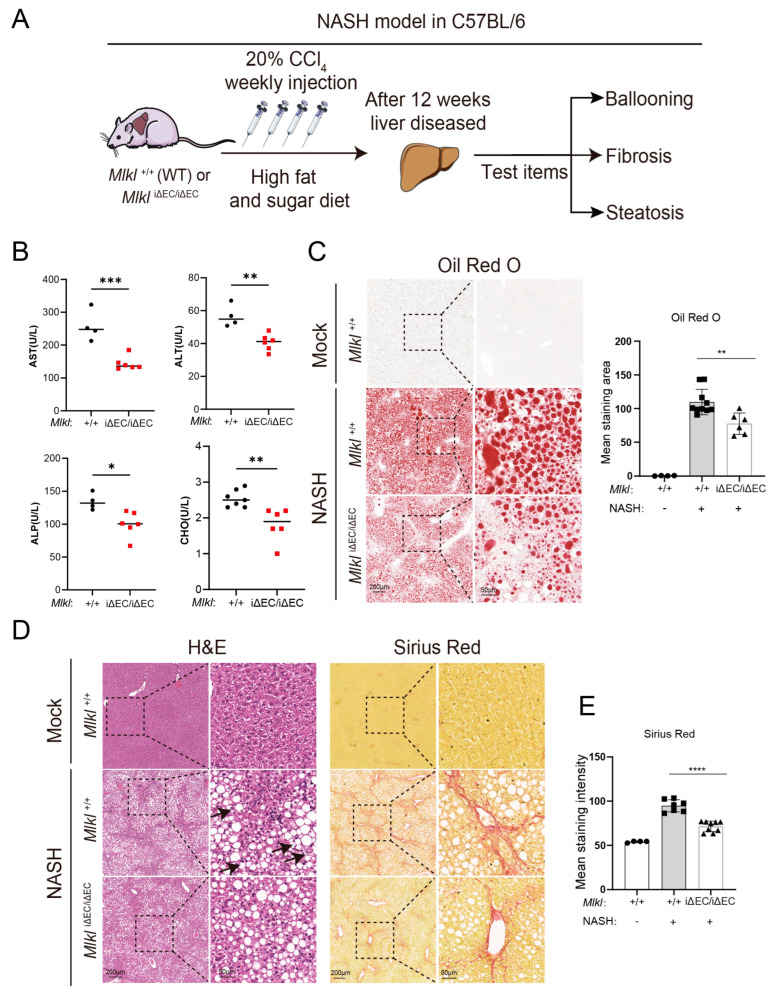
Deletion of endothelial MLKL alleviates histopathological phenotype associated with NASH progression in mice. (**A**) Schema depicting approach to build a mouse NASH model and the indicators to describe progression of NASH. (**B**) Concentrations of serum AST, ALT, ALP, and CHO in *Mlkl*^+/+^ and *Mlkl^i^*^ΔEC/iΔEC^ NASH model. *n* ≥ 4 per group. AST: aspartate amino transferase, ALT: alanine aminotransferase, ALP: alkaline phosphatase, CHO: cholesterol. (**C**) Oil Red O staining was carried out and the result was quantified in *Mlkl*^+/+^ and *Mlkl^i^*^ΔEC/iΔEC^ mice with corresponding groups. Left: ROI (region of interest, 8×), scale bars: 200 μm. Right: Magnified images of indicated dotted box areas (40×), scale bars: 50 μm. (**D**) Liver histopathology analyzed by H&E, Sirius Red staining, and quantification (**E**) of *Mlkl*^+/+^ mice and *Mlkl^i^*^ΔEC/iΔEC^ mice with corresponding groups. Left: ROI (region of interest, 8×), scale bars: 200 μm. Right: Magnified images of indicated dotted box areas (40×), scale bars: 50 μm. Black arrow: Ballooned hepatocytes. *n* ≥ 4 per group. Data are shown as means ± SEMs. * *p* < 0.05, ** *p* < 0.01, *** *p* < 0.001, **** *p* < 0.0001.

**Figure 3 ijms-24-11313-f003:**
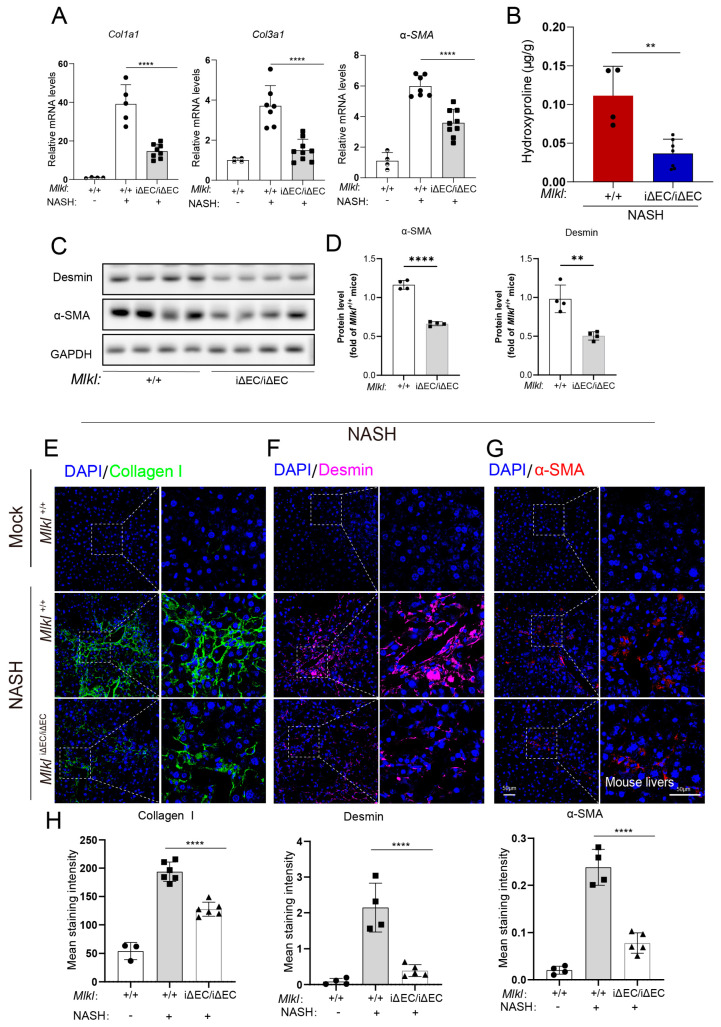
Deletion of endothelial MLKL mitigates fibrosis in NASH model. (**A**) Relative mRNA level of *Col1a1*, *Col3a1*, and *α*-*SMA* in liver tissue were quantified by RT-qPCR showing there was a reduction in *Mlkl*^iΔEC/iΔEC^ mice compared with *Mlkl*^+/+^ mice in NASH model. *n* ≥ 4 per group. (**B**) Hydroxyproline assay was assessed in *Mlkl*^+/+^ and *Mlkl^i^*^ΔEC/iΔEC^ NASH model. *n* ≥ 4 per group. (**C**) Protein levels of Desmin, α-SMA in *Mlkl*^+/+^, and *Mlkl^i^*^ΔEC/iΔEC^ NASH model was shown by Western blot and quantified (**D**). (**E**–**G**) Immunostaining for Collagen I (green), Desmin (purple), α-SMA (red), and DAPI (blue, nucleus) in *Mlkl*^+/+^ and *Mlkl^i^*^ΔEC/iΔEC^ NASH model. Left: ROI (region of interest), scale bars: 50 μm. Right: Magnified images of indicated dotted box areas, scale bars: 50 μm. (**H**) The mean staining intensity were quantified by image J (**H**). *n* ≥ 4 per group. Data are shown as means ± SEMs. ** *p* < 0.01, **** *p* < 0.0001.

**Figure 4 ijms-24-11313-f004:**
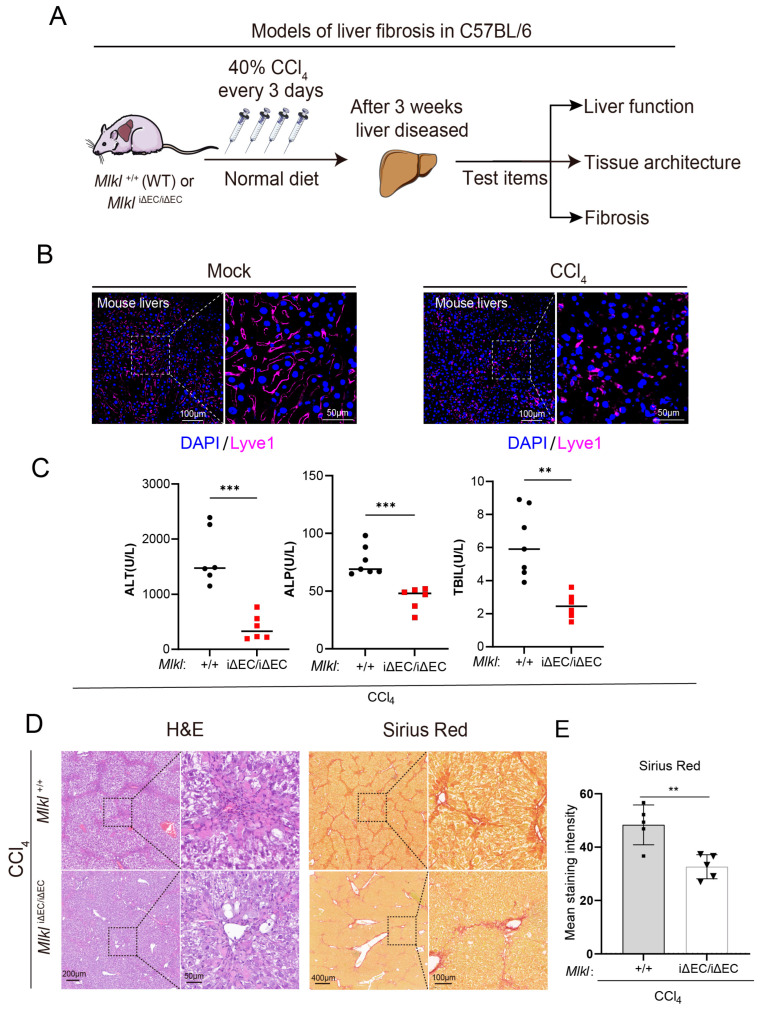
Deficiency of endothelial MLKL improves the liver function and mitigates inflammation and fibrosis in CCl_4_ model. (**A**) Approach to generating a liver fibrosis model in mice. (**B**) Representative images of immunostaining for Lyve1 (purple) in liver sections. Left: ROI (region of interest, 20×), scale bars: 100 μm. Right: Magnified images of indicated dotted box areas, scale bars: 50 μm. (**C**) Serum content of liver function index including ALT, ALP, and TBIL in *Mlkl*^+/+^ and *Mlkl^i^*^ΔEC/iΔEC^ fibrosis model. ALT: alanine aminotransferase, ALP: alkaline phosphatase, total bilirubin. *n* ≥ 3 per group. (**D**) H&E and Sirius Red staining in *Mlkl*^+/+^ and *Mlkl^i^*^ΔEC/iΔEC^ mice after CCl4 injection. Left: ROI (region of interest, 8×). Right: Magnified images of indicated dotted box areas (40×), scale bars as shown in pictures. (**E**) Quantification of Sirius Red staining shown. *n* ≥ 5 per group. Data are shown as means ± SEMs. ** *p* < 0.01, *** *p* < 0.001.

**Figure 5 ijms-24-11313-f005:**
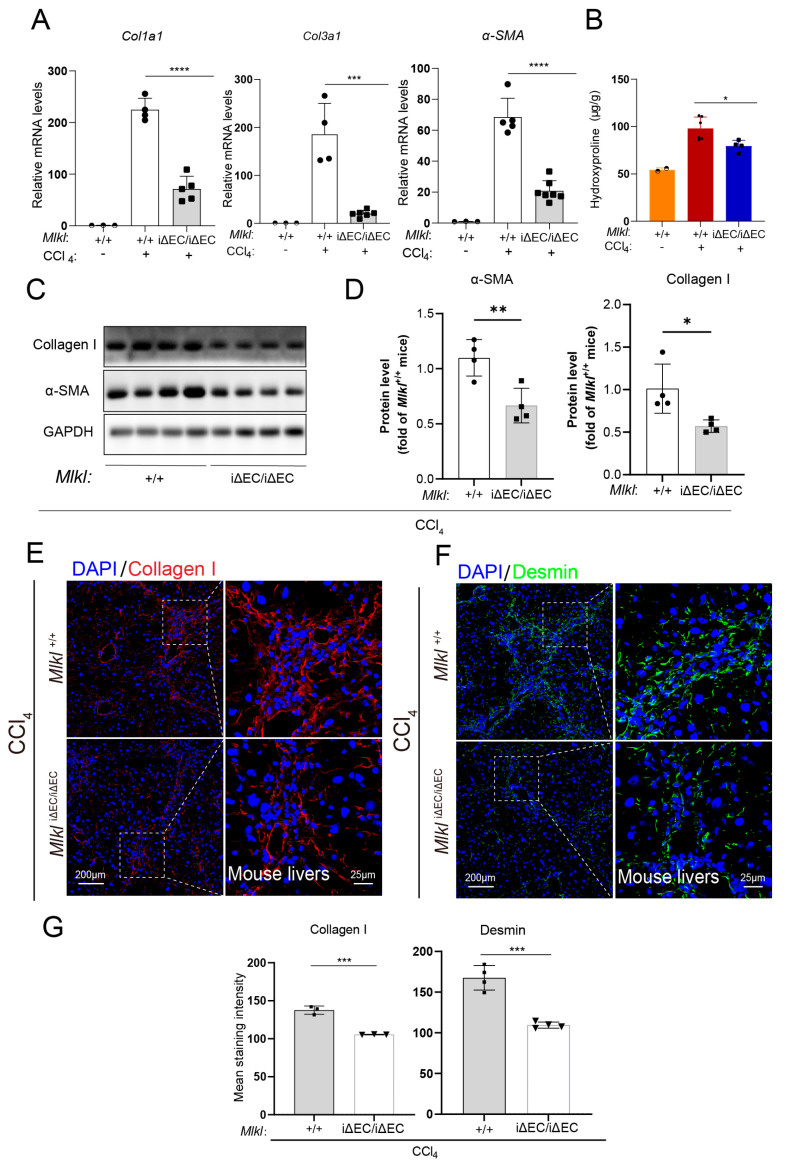
Deficiency of endothelial MLKL downregulates the liver fibrosis indexes in CCl_4_ model. (**A**) RT-qPCR of *Col1a1*, *Col3a1*, and *α*-*SMA* expression in liver tissues of mock and fibrosis mice. *n* ≥ 3 per group. (**B**) Liver hydroxyproline amounts were detected in control versus *Mlkl*^+/+^ and *Mlkl^i^*^ΔEC/iΔEC^ mice of NASH model. *n* ≥ 3 per group. (**C**) α-SMA and Collagen I protein abundance in liver tissues of indicated mouse groups in the NASH model as indicated by Western blot and quantification (**D**). *n* ≥ 3 per group. (**E**,**F**) Immunostaining for Collagen I (red), Desmin (green), and DAPI (blue) in *Mlkl*^+/+^ and *Mlkl^i^*^ΔEC/iΔEC^ mice that were subjected to CCl_4_ injections. Left: ROI (region of interest), scale bars, 200 μm. Right: Magnified images of indicated dotted box areas, scale bars: 25 μm. (**G**) Mean staining was quantified by image J. *n* ≥ 3 per group. Data are shown as means ± SEMs. * *p* < 0.05, ** *p* < 0.01, *** *p* < 0.001, **** *p* < 0.0001.

**Figure 6 ijms-24-11313-f006:**
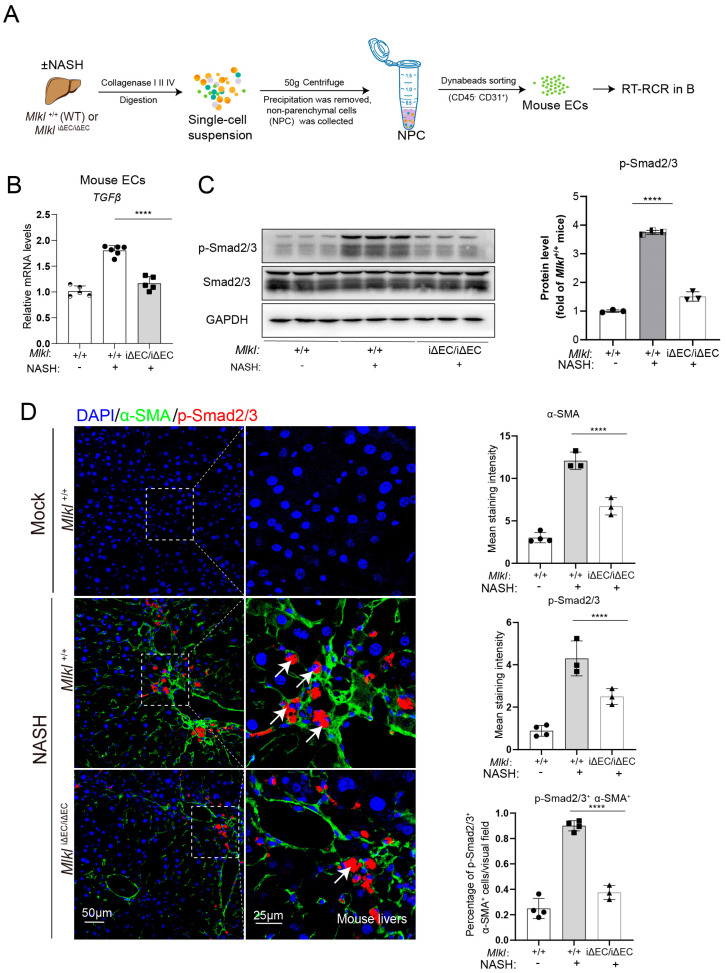
EC-specific knockout of Mlkl reduces the pro-fibrotic TGFβ/Smad 2/3 signaling axis between endothelial cells and HSCs. (**A**) The experimental scheme for isolating primary endothelial cells (ECs) from livers of indicated mice. (**B**) RT-qPCR analyses of TGFβ in mouse ECs from Mlkl^+/+^ and Mlkl^iΔEC/iΔEC^ NASH model. (**C**) Immunoblotting analyses of Smad2/3 and p-Smad2/3 expression in the mouse liver. (**D**) Representative immunostaining images shown that the expression of p-Smad2/3 (red) colocalized with activated HSCs marker, α-SMA (green) in liver from Mlkl^+/+^, and Mlkl^iΔEC/iΔEC^ NASH model. Activated HSCs co-located with p-Smad2/3 (White arrows). DAPI staining was used to identify the nuclei. Summarized histogram shows the average area of α-SMA, p-Smad2/3, and co-localization of p-Smad2/3 with α-SMA per field. Left: ROI (region of interest), scale bars: 50 μm. Right: Magnified images of indicated dotted box areas, scale bars: 25 μm. *n* ≥ 3 per group. Data are shown as means ± SEMs. **** *p* < 0.0001.

**Figure 7 ijms-24-11313-f007:**
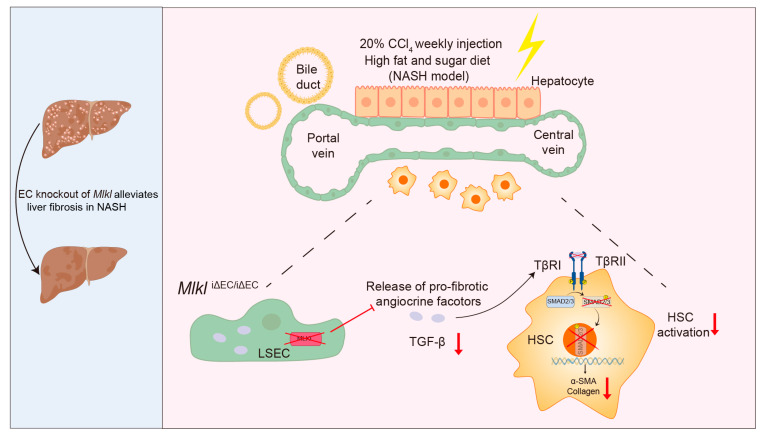
Proposed model illustrating that EC-specific knockout of *Mlkl* (*Mlkl*^iΔEC/iΔEC^) alleviates liver fibrosis in NASH by mitigating the activation of TGFβ/Smad2/3 pathway and disrupting the pro-fibrotic crosstalk between endothelial cells (LSEC)–hepatic stellate cells (HSCs). LSEC: liver sinusoidal endothelial cells, TGF-β: transforming growth factor beta, TβRI: TGF-β receptor I, TβR II: TGF-β receptor II, HSC: hepatic stellate cell.

## Data Availability

The data presented in the study are available in the article.

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
