# Peer review of "Targeting Endothelial Necroptosis Disrupts Profibrotic Endothelial–Hepatic Stellate Cells Crosstalk to Alleviate Liver Fibrosis in Nonalcoholic Steatohepatitis"

_ijms, 2023, doi:10.3390/ijms241411313_

Round 1

Reviewer 1 Report

In this work, the authors show that SEC-specific knock-out of MLKL resulted in an improvement of fibrosis in two different murine models. However, no attempt was made to elucidate or hypothesize the mechanism, even though changes in various fibrosis markers were evaluated in control and SEC-specific MLKL knock-out mice.

Major requests:

1)      Some attempt to elucidate the mechanism must be done, for example by assaying the TGFβ/Smad 2/3 pathway activation state involved in HSC activation, or maybe the activation/expression of factors involved in fibrosis, i.e. ADAM, RECK and MMP9.

2)      An hypothesis about the mechanism should be made, also based on the added results (see point 1)

3)      The discussion should be more detailed; in particular, a paragraph about the role of SECs in the development of NASH and fibrosis, in view of the obtained results, is completely lacking and should be added

4)      In addition, the findings here presented should be compared with those obtained by Guo et al, 2022, in order to better focus on the role of SECs in comparison with hepatocytes.

Minor observations:

1)      The English should be thoroughly revised

2)      Line 83-85: citation is needed

3)      Lines 111-127: the method is not described clearly enough. The process should be made understandable also to people not familiar to this technique. Citations are also needed.

English needs revision.

Reviewer 2 Report

"Blockade of endothelial necroptosis alleviates fibrosis in nonalcoholic steatohepatitis" have been written by Yan M & Li H et al. They study the effect of necroptosis of liver sinusoidal endothelial cells on the injured liver in 2 mouse models (One is the NASH model and the other CCl4 model).  

1. Introduction

What is the status of NAFLD in China? The authors are from China and Chinese given Data from this country makes sense.

2. Results

-line 248: authors indicated " inflammation (...) hepatocytes" this is not shown in figure 2. 

3. Figures

- Figure 1B: Bands from F1/R1 PCR should be improved

- Figure 2C: a histogram with Oil Red O quantification should be added

- Figure 2D red Sirius pictures in low magnification in MLKL iDEC are so clear giving the impression to hide the red staining.

- Figure 3E/Figure 4B/Figure 5B: difficulties to see the DAPI staining on printed paper

- Figure 4D: Sirius Red pictures should be improved

Round 2

Reviewer 1 Report

I thank the authors for considering my recommendations. I agree with the publication of the manuscript in the revised form.